

# BGDMdocker: a Docker workflow for data mining and visualization of bacterial pan-genomes and biosynthetic gene clusters

Gong Cheng[1,2], Quan Lu[3], Ling Ma[4], Guocai Zhang[4], Liang Xu[5] and Zongshan Zhou[1]

[1] Protection Research Center of Pomology, Research Institute of Pomology, Chinese Academy of Agricultural Sciences, Xingcheng, China
[2] Forest Protection Research Institute of Heilongjiang Province, Harbin, China
[3] Research Institute of Forest Ecology, Environment and Protection, Chinese Academy of Forestry, Beijing, China
[4] School of Forestry, Northeast Forestry University, Harbin, China
[5] Institute of Food Science and Technology, Chinese Academy of Agricultural Sciences, Beijing, China

## ABSTRACT

Recently, Docker technology has received increasing attention throughout the bioinformatics community. However, its implementation has not yet been mastered by most biologists; accordingly, its application in biological research has been limited. In order to popularize this technology in the field of bioinformatics and to promote the use of publicly available bioinformatics tools, such as Dockerfiles and Images from communities, government sources, and private owners in the Docker Hub Registry and other Docker-based resources, we introduce here a complete and accurate bioinformatics workflow based on Docker. The present workflow enables analysis and visualization of pan-genomes and biosynthetic gene clusters of bacteria. This provides a new solution for bioinformatics mining of big data from various publicly available biological databases. The present step-by-step guide creates an integrative workflow through a Dockerfile to allow researchers to build their own Image and run Container easily.

**Submitted** 21 June 2017
**Accepted** 30 September 2017
**Published** 30 November 2017

Corresponding authorss
Quan Lu, luquan@caf.ac.cn
Ling Ma, maling63@163.com
Zongshan Zhou, zszhouqrj@163.com

## INTRODUCTION

Docker is an open source project and platform for building, shipping, and running any app, enabling the widespread distribution of applications. Docker allows users to package an application, along with all its dependencies, into a standardized unit for software development (https://docs.docker.com/). Docker includes three core structural compositions: Image, Container, and Repository. An image can start software as complex as a database, wait for you (or someone else) to add data, store the data for later use, and then wait for the next person. Containers afford similar resource isolation and allocation benefits as virtual machines; however, a different architectural approach allows the former

to be much more portable and efficient. When an app is in Docker containers, setting up and maintaining different environments or tools for each language is not necessary (https://hub.docker.com/). The Docker Hub Registry allows users to find, manage, and pull Images from community, official, and private image libraries, and is free to use for public repositories (https://www.docker.com/whatisdocker). GitHub is a web-based source code version control repository and Internet hosting service that is mostly used for code. Compared with GitHub, Docker Hub (https://hub.docker.com/) is a cloud-based registry service of Docker, whose most notable advantages include workflow automation throughout the development pipeline based on Images and Counter of Docker.

Academic bioinformatics software programs generally suffer from limitations such as installation and configuration difficulties, large dependencies, and restrictions on the amount of data that may be uploaded to online servers. Therefore, several excellent software programs are of limited use to biologists. Bioinformatics tools may be merged with Docker technology to build reproducible and convenient types of workflows. Docker provides programmers, development teams, and bioinformaticians with a common toolbox that allows users to take full advantage of bioinformatics tools, thus helping to build, ship, and run any app, as well as distribute apps anywhere. Docker technology is suitable for use in the field of bioinformatics because of certain advantages and characteristics that allow applications to run in an isolated, self-contained package. This package may be efficiently distributed and executed in a portable manner across a wide range of computing platforms (*Aranguren & Wilkinson, 2015*; *Belmann et al., 2015*; *Hosny et al., 2016*). To date, numerous bioinformatics tools based on Docker have been developed and published in different programming languages such as Perl and BioPerl (https://hub.docker.com/_/perl/; https://hub.docker.com/r/bioperl/bioperl/), python and biopython (https://hub.docker.com/_/python/; https://hub.docker.com/r/biopython/biopython), and R and Bioconductor (https://hub.docker.com/_/r-base/; https://hub.docker.com/r/bioconductor/release_base/). These projects have contributed to official Docker Images. The famous Galaxy program has also contributed to Docker Galaxy (*Grüning, 2017*).

Here, we used Docker technology to rapidly construct a pan-genome analysis process that may be used in Linux, Windows, or Mac environments (64-bit). The present process may also be deployed as a cloud-based system such as with Amazon EC2 or other cloud providers. This workflow should provide a useful service to biologists in the field of bioinformatics. Docker Containers have only a minor impact on the performance of common genomic pipelines (*Tommaso et al., 2015*).

*Bacillus amyloliquefaciens* has been extensively studied as an important biological control agent owing to its ability to inhibit the growth of fungi and bacteria (*Nam et al., 2016*). Using Docker, we rapidly executed a container (on Ubuntu 16.04 and Win10 hosts) to analyze the pan-genome and reveal biosynthetic gene cluster features of 44 *B. amyloliquefaciens* strains, as well as to visualize the results. The analytical workflow consisted of three toolkits: Prokka v1.11 (*Seemann, 2014*), panX (*Ding, Baumdicker & Neher, 2017*), and antiSMASH3.0 (*Weber et al., 2015*), for prokaryotic genome annotation, pan-genome analysis and visualization, and analysis of biosynthetic gene clusters, respectively.

We included all of these applications and their dependencies in a BGDMdocker (bacterial genome data mining Docker-based) to enable the workflow to be implemented online with a single run. We additionally wrote three standalone Dockerfiles for Prokka, panX, and antiSMASH in order to meet the various requirements of different users. We recommend setting up the workflow with three independent files, each with a specific purpose. This method is presented in the Supplementary Information. Here, we describe how to build the workflow and conduct the analysis in detail.

## MATERIALS AND METHODS

### Installation of latest Docker on your host

1. Copy the following commands for quickly and easily installing the latest Docker-CE (https://docs.docker.com/engine/installation/) (Ubuntu, Debian, Raspbian, Fedora, Centos, Redhat, Suse, Oracle, Linux etc. are all applicable):

```
$ curl -fsSL get.docker.com -o get-docker.sh
$ sudo sh get-docker.sh
```

If user would like to use Docker as a non-root user, you should now consider adding your user to the "docker" group, e.g., using:

```
$ sudo usermod -aG docker <user name>
```

Type the following commands at your shell prompt. If this outputs the Docker version, your installation was successful.

```
$ docker version
```

2. Installing latest Docker on Windows 10 Enterprise (https://docs.docker.com/docker-for-windows/install/):

The current version of Docker for Windows runs on 64-bit Windows 10 Pro, Enterprise, and Education editions. Download "InstallDocker.msi" (https://download.docker.com/win/stable/InstallDocker.msi). Double-click "InstallDocker.msi" to run the installer. Follow the install wizard to accept the license, authorize the installer, and proceed with the installation.

Type the following commands at your shell prompt (cmd.exe or PowerShell). If this outputs the Docker version, your installation was successful.

```
$ docker version
```

### Use Docker to build the BGDMdocker workflow

1. On your host (with Docker), type the following command lines to build a BGDMdocker workflow:

```
$ git clone https://github.com/cgwyx/BGDMdocker.git
```

or: download "BGDMdocker-master.zip" (https://github.com/cgwyx/BGDMdocker/archive/master.zip) file

```
$ unzip BGDMdocker-master.zip
```

2. Build workflow Images:

```
$ cd ./BGDMdocker
$ docker build -t BGDMdocker:latest .
```

Or: pull Images of BGDMdocker from DockerHub (https://hub.docker.com/r/cgwyx/bgdmdocker/), such as:

```
$ docker pull cgwyx/bgdmdocker
```

3. Run a Container from the BGDMdocker Image:

```
$ docker run -it --rm -v home:home -p 8000:8000 BGDMdocker:latest
/bin/bash
```

If use the "-v home:home" parameter, Docker will mount the local folder/home into the Container under /home, storing all of your data in one directory in the home folder of the host operating system; then, you may access the directories of home from inside the Container.

We analyzed the pan-genome and biosynthetic gene clusters of 44 *B. amyloliquefaciens* strains using the BGDMdocker workflow. For detailed commands, see the Supplementary Information.

## RESULTS

### Fast and reproducible building of the BGDMdocker workflow across computing platforms using Docker

Using Docker technology, the Dockerfile script file can build Images and run a container in seconds or milliseconds on Linux and Windows. The file may also be deployed in Mac and cloud-based systems such as Amazon EC2 or other cloud providers. The Dockerfile is a small, plain-text file that may be easily stored and shared. Therefore, the user is not required to install and configure the programs.

Here, based on Debian 8.0 (Jessie) Image, we have established a novel Docker-based bioinformatics platform for the study of microbe genomes and pan-genomes (Fig. 1). The workflow, which offers the advantages of cross-platform and modular reuse, provides biologists with simple and standardized tools to extract biological information from their own experiments and from online sequence databases. Researchers may therefore focus solely on mining information from the obtained sequences rather than determining how to install the software package. We have uploaded this Dockerfile to GitHub for sharing with relevant scientific researchers.

### Datamining and visualizing the pan-genomes of *B. amyloliquefaciens*

In order to explore the result data, a website (http://bapgd.hygenomics.com/pangenome/home) was built for the interactive exploration of the *B. amyloliquefaciens* pan-genome and biosynthetic gene clusters using the BGDMdocker workflow. Visualization allowed

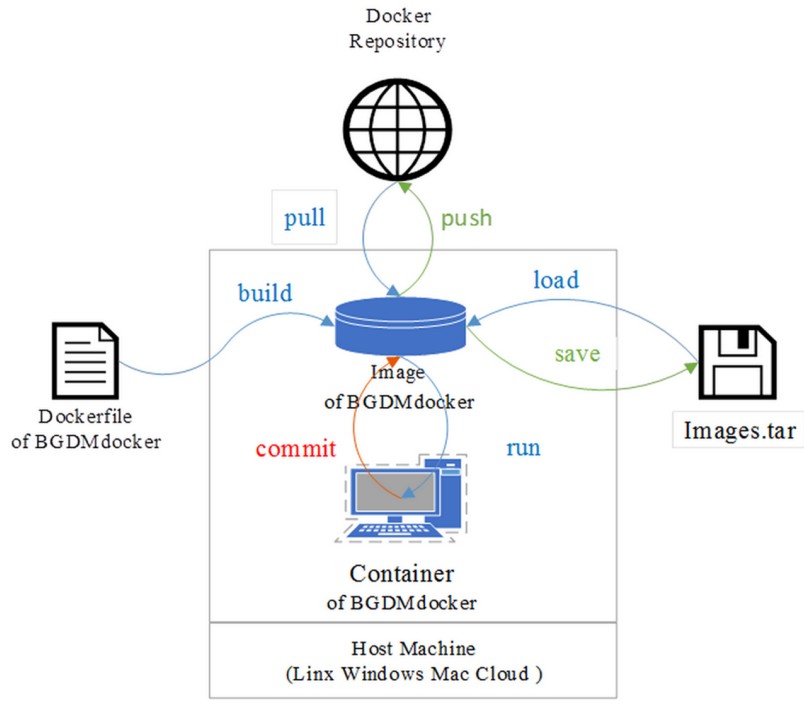

**Figure 1 Schematic of BGDMdocker workflow based on Docker: building image and running container from Dockerfile, then login interaction patterns to run software.** This enables the user to load and save Image.tar, run and commit Container, pull from and push to Docker repository.

for the rapid filtering and searching of genes. For each gene cluster, panX displayed an alignment and a phylogenetic tree, mapped mutations within that cluster to the branches of the tree, and inferred gene losses and gains on the core-genome phylogeny. Here, we provide the summary statistics of the pan-genome (Table 1), the phylogenetic relationships of the 44 *B. amyloliquefaciens* strains (Fig. 2), and screenshots of the website (http://bapgd.hygenomics.com/pangenome/home) (Figs. 3 and 4). All data may be visualized and downloaded without registration.

## Datamining and visualizing of biosynthetic gene clusters of *B. amyloliquefaciens*

Results from the identification and analysis of the biosynthetic gene clusters of 44 *B. amyloliquefaciens* strain genomes, using the BGDMdocker workflow, have been uploaded to our website (http://bapgd.hygenomics.com/pangenome/home). All data may be downloaded without registration.

Here, we provide brief summary statistics for the biosynthetic gene clusters of all 44 strains (Table 2), as well as an example of the type and number of biosynthetic gene clusters in the Y2 (NC_017912) strain (Table 3) and representative screenshots of the website (http://bapgd.hygenomics.com/pangenome/home) (Figs. 5 and 6). There are a total of 31 gene clusters in the genome of Y2. Among these, 21 gene clusters show similarities to known clusters in MIBiG (http://mibig.secondarymetabolites.org/)

**Table 1 Summary statistics of pan-genome of 44 *B. amyloliquefaciens* strains.**

| Accession | Strains | Gene numbers in pan-genome of *B. amyloliquefaciens* (total genes, 172,388; core gene clusters, 2,306) | | | | Gene of strain genomes | |
|---|---|---|---|---|---|---|---|
| | | Total gene | Core gene | Acc gene | Uni gene | All gene | All protein |
| CYHL01000001 | JRS5 | 3,856 | 2,310 | 1,546 | 57 | 3,870 | 3,863 |
| CYHP01000001 | JRS8 | 3,994 | 2,311 | 1,683 | 118 | 4,016 | 4,006 |
| NC_014551 | DSM7 | 3,935 | 2,307 | 1,628 | 21 | 4,030 | 3,811 |
| NC_017188 | TA208 | 3,935 | 2,307 | 1,628 | 1 | 3,974 | 3,847 |
| NC_017190 | LL3 | 3,981 | 2,308 | 1,673 | 19 | 4,037 | 3,887 |
| NC_017191 | XH7 | 3,942 | 2,307 | 1,635 | 6 | 3,983 | 3,846 |
| NC_017912 | Y2 | 4,099 | 2,310 | 1,789 | 46 | 4,148 | 3,983 |
| NC_020272 | IT-45 | 3,803 | 2,310 | 1,493 | 4 | 3,832 | 3,678 |
| NC_022653 | CC178 | 3,754 | 2,310 | 1,444 | 19 | 3,795 | 3,641 |
| NC_023073 | LFB112 | 3,761 | 2,308 | 1,453 | 19 | 3,801 | 3,637 |
| NZ_AUNG01000001 | Lx-11 | 3,700 | 2,309 | 1,391 | 5 | 3,742 | 3,619 |
| NZ_AUWK01000001 | HB-26 | 3,797 | 2,311 | 1,486 | 30 | 3,842 | 3,714 |
| NZ_AVQH01000001 | EGD-AQ141 | 4,079 | 2,311 | 1,768 | 54 | 4,121 | 3,995 |
| NZ_AWQY01000001 | UASWS BA1 | 3,794 | 2,309 | 1,485 | 8 | 3,806 | 3,681 |
| NZ_CP006058 | UMAF6639 | 3,825 | 2,311 | 1,514 | 20 | 3,879 | 3,716 |
| NZ_CP006960 | UMAF6614 | 3,804 | 2,311 | 1,493 | 13 | 3,850 | 3,695 |
| NZ_CP007242 | KHG19 | 3,775 | 2,310 | 1,465 | 19 | 3,816 | 3,658 |
| NZ_CP010556 | L-H15 | 3,724 | 2,309 | 1,415 | 6 | 3,769 | 3,615 |
| NZ_CP011278 | L-S60 | 3,728 | 2,310 | 1,418 | 7 | 3,773 | 3,611 |
| NZ_CP013727 | MBE1283 | 3,794 | 2,314 | 1,480 | 24 | 3,856 | 3,681 |
| NZ_CP014700 | S499 | 3,776 | 2,310 | 1,466 | 5 | 3,819 | 3,671 |
| NZ_CP014783 | B15 | 3,820 | 2,315 | 1,505 | 13 | 3,875 | 3,704 |
| NZ_CP016913 | RD7-7 | 3,597 | 2,308 | 1,289 | 39 | 3,656 | 3,483 |
| NZ_DF836091 | CMW1 | 3,771 | 2,311 | 1,460 | 128 | 3,901 | 3,706 |
| NZ_JCOC01000001 | EBL11 | 3,733 | 2,308 | 1,425 | 20 | 3,773 | 3,682 |
| NZ_JMEG01000001 | B1895 | 3,824 | 2,306 | 1,518 | 167 | 4,026 | 3,623 |
| NZ_JQNZ01000001 | X1 | 3,724 | 2,309 | 1,415 | 3 | 3,766 | 3,619 |
| NZ_JTJG01000001 | JJC33M | 3,888 | 2,309 | 1,579 | 121 | 3,952 | 3,796 |
| NZ_JXAT01000001 | LPL-K103 | 3,709 | 2,309 | 1,400 | 15 | 3,743 | 3,637 |
| NZ_JZDI01000001 | 12B | 8,166 | 2,354 | 5,812 | 4,040 | 8,194 | 7,985 |
| NZ_KB206086 | DC-12 | 3,910 | 2,311 | 1,599 | 50 | 3,984 | 3,842 |
| NZ_KN723307 | TF281 | 3,640 | 2,312 | 1,328 | 5 | 3,782 | 3,571 |
| NZ_LGYP01000001 | 629 | 3,536 | 2,313 | 1,223 | 11 | 3,785 | 3,427 |
| NZ_LJAU01000001 | Bs006 | 4,042 | 2,312 | 1,730 | 46 | 4,074 | 3,969 |
| NZ_LJDI01000020 | XK-4-1 | 3,799 | 2,310 | 1,489 | 14 | 3,821 | 3,701 |
| NZ_LMAG01000001 | RHNK22 | 3,781 | 2,309 | 1,472 | 37 | 3,837 | 3,698 |
| NZ_LMAT01000001 | Jxnuwx-1 | 3,930 | 2,309 | 1,621 | 246 | 4,008 | 3,870 |
| NZ_LMUC01000016 | H57 | 3,816 | 2,310 | 1,506 | 42 | 3,859 | 3,732 |
| NZ_LPUP01000011 | 11B91 | 3,790 | 2,311 | 1,479 | 49 | 3,892 | 3,702 |

| Accession | Strains | Gene numbers in pan-genome of *B. amyloliquefaciens* (total genes, 172,388; core gene clusters, 2,306) | | | | Gene of strain genomes | |
|---|---|---|---|---|---|---|---|
| | | Total gene | Core gene | Acc gene | Uni gene | All gene | All protein |
| NZ_LQQW01000001 | M49 | 3,694 | 2,311 | 1,383 | 21 | 3,741 | 3,617 |
| NZ_LQYO01000001 | B4140 | 3,771 | 2,307 | 1,464 | 49 | 3,847 | 3,713 |
| NZ_LQYP01000001 | B425 | 3,921 | 2,310 | 1,611 | 39 | 4,034 | 3,844 |
| NZ_LYUG01000001 | SRCM101266 | 3,724 | 2,306 | 1,418 | 15 | 3,781 | 3,628 |
| NZ_LZZO01000001 | SRCM101294 | 3,946 | 2,308 | 1,638 | 175 | 3,982 | 3,850 |

**Note:**
Genome sequences of 44 *B. amyloliquefaciens* (https://www.ncbi.nlm.nih.gov/genome/genomes/848) strains downloaded from GenBank RefSeq database: "Acc gene" refers to accessory gene (dispensable gene); "Uni gene" refers to unique gene (strain-specific gene); "All genes" refers to gene of *.gbff files recorder, including Pseudo Genes; "Total genes" refers to those used for pan-genome analysis gene of *.gbff files recorder, excluding Pseudo Genes.

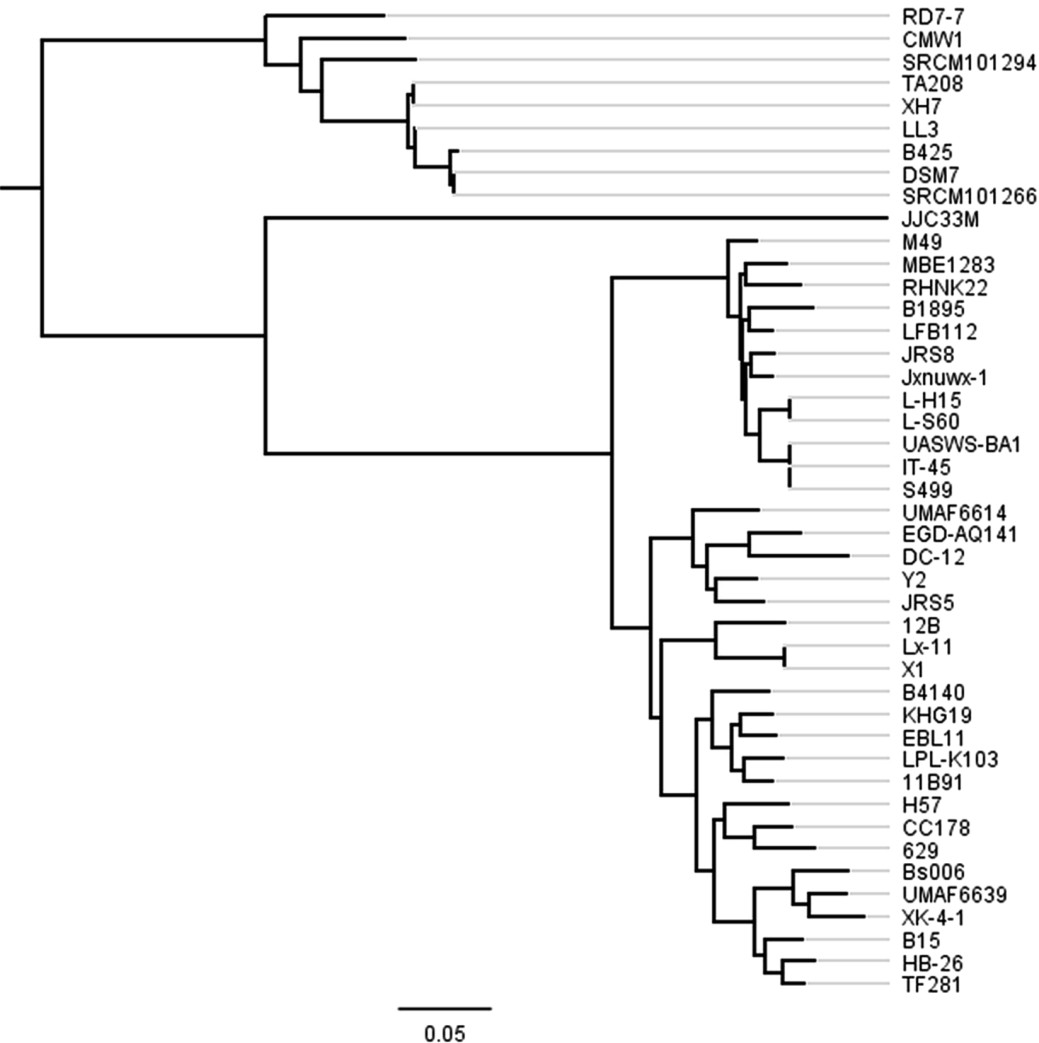

**Figure 2 Phylogenetic tree of 44 *B. amyloliquefaciens* strains.** The tree was constructed using all genes shared between the 44 strains (2,306 core genes). The scale bar represents genetic distance.

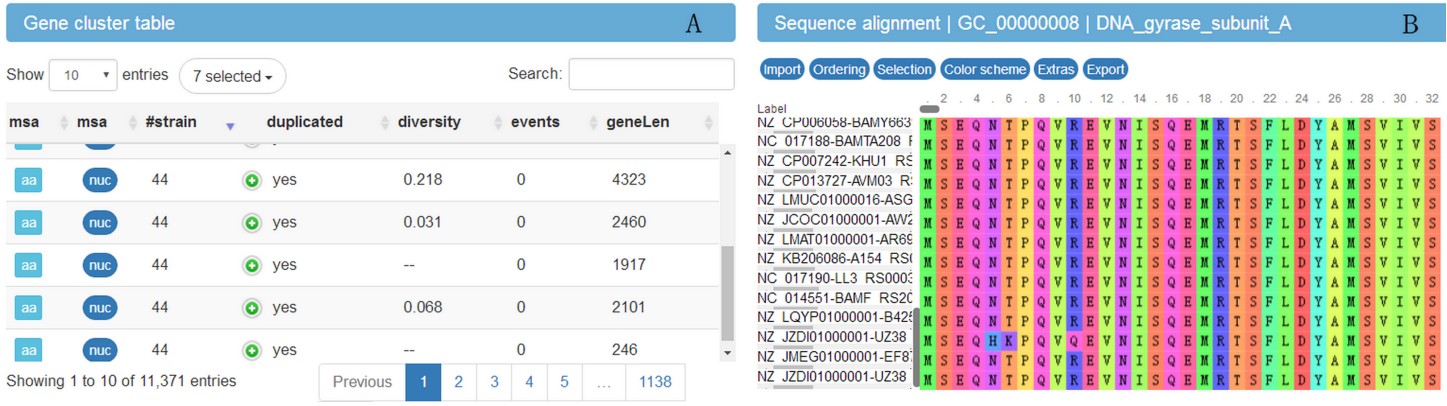

**Figure 3 Screenshot of website (http://bapgd.hygenomics.com/pangenome/home) for visualization of gene cluster table (A) and sequence alignment (B).**

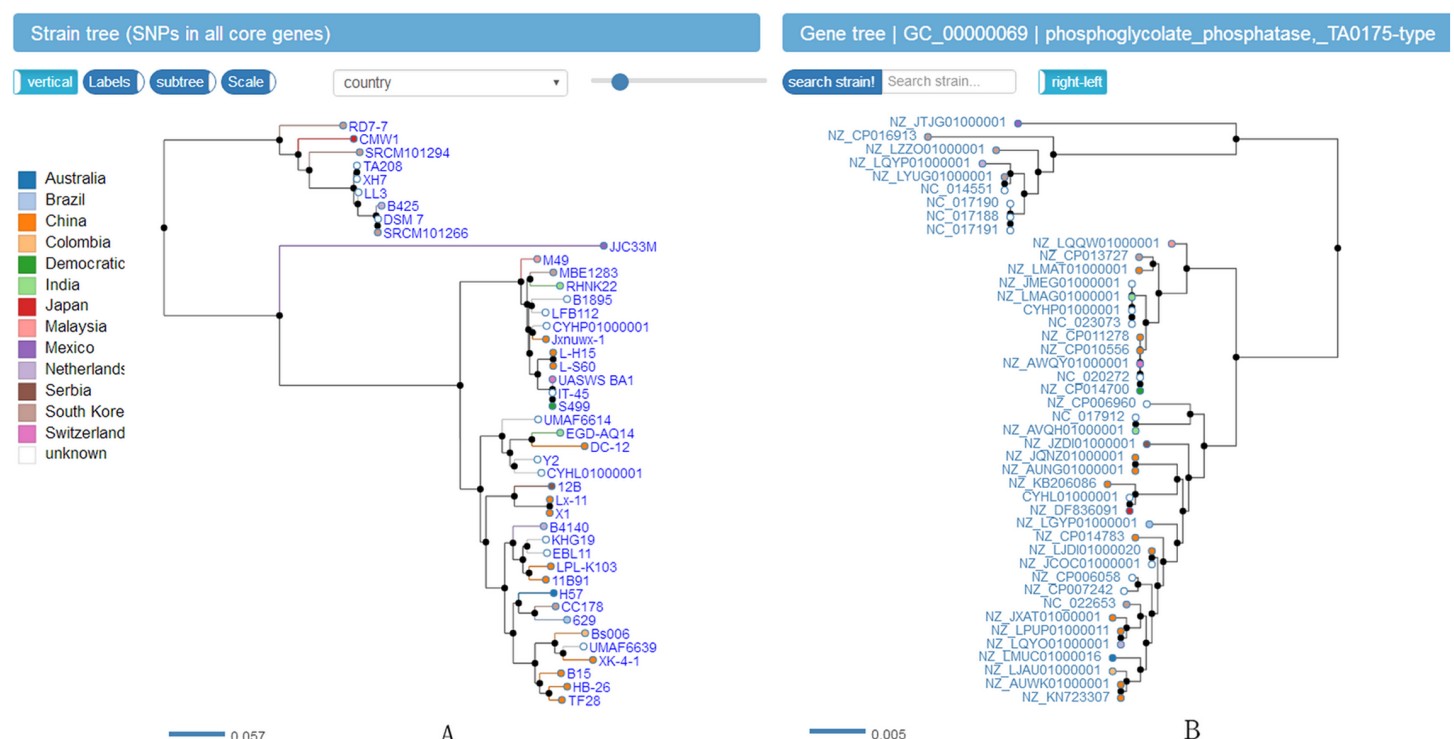

**Figure 4 Screenshot of website (http://bapgd.hygenomics.com/pangenome/home) for visualization of Phylogenetic tree of 44 *B. amyloliquefaciens* strains (A) and genes (B). The scale bar represents genetic distance.**

such as surfactin, mersacidin, and fengycin; the remaining 10 gene clusters are unknown.

## DISCUSSION

The Dockerfiles of BGDMdocker scripts are convenient for deployment and sharing, and it is easy for other users to customize the Images by editing the Dockerfile directly. This is in contrast to Makefiles and other installations, for which the resulting builds differ

**Table 2 Summary statistics of biosynthetic gene clusters of 44 B. amyloliquefaciens strains.**

| Accession | Strains | Biosynthetic gene clusters | | | | Genome of strains | | |
|---|---|---|---|---|---|---|---|---|
| | | Total | Known | Unknown | Type | Size (Mb) | Gene | Protein |
| CYHL01000001 | JRS5 | 38 | 27 | 11 | 12 | 4.03148 | 3,870 | 3,863 |
| CYHP01000001 | JRS8 | 42 | 26 | 16 | 11 | 4.0909 | 4,016 | 4,006 |
| NC_014551 | DSM7 | 31 | 18 | 13 | 9 | 3.9802 | 4,030 | 3,811 |
| NC_017188 | TA208 | 29 | 17 | 12 | 10 | 3.93751 | 3,974 | 3,847 |
| NC_017190 | LL3 | 29 | 17 | 12 | 9 | 4.00199 | 4,037 | 3,887 |
| NC_017191 | XH7 | 29 | 17 | 12 | 10 | 3.9392 | 3,983 | 3,846 |
| NC_017912 | Y2 | 31 | 21 | 10 | 11 | 4.23862 | 4,148 | 3,983 |
| NC_020272 | IT-45 | 32 | 19 | 13 | 11 | 3.93687 | 3,832 | 3,678 |
| NC_022653 | CC178 | 35 | 19 | 16 | 9 | 3.91683 | 3,795 | 3,641 |
| NC_023073 | LFB112 | 35 | 20 | 15 | 10 | 3.94275 | 3,801 | 3,637 |
| NZ_AUNG01000001 | Lx-11 | 37 | 25 | 12 | 11 | 3.88689 | 3,742 | 3,619 |
| NZ_AUWK01000001 | HB-26 | 45 | 30 | 15 | 9 | 3.98936 | 3,842 | 3,714 |
| NZ_AVQH01000001 | EGD-AQ141 | 36 | 26 | 10 | 12 | 4.22259 | 4,121 | 3,995 |
| NZ_AWQY01000001 | UASWS BA1 | 37 | 25 | 12 | 11 | 3.94409 | 3,806 | 3,681 |
| NZ_CP006058 | UMAF6639 | 35 | 21 | 14 | 10 | 4.03464 | 3,879 | 3,716 |
| NZ_CP006960 | UMAF6614 | 32 | 20 | 12 | 10 | 4.00514 | 3,850 | 3,695 |
| NZ_CP007242 | KHG19 | 32 | 20 | 12 | 10 | 3.95336 | 3,816 | 3,658 |
| NZ_CP010556 | L-H15 | 32 | 19 | 13 | 10 | 3.90597 | 3,769 | 3,615 |
| NZ_CP011278 | L-S60 | 32 | 19 | 13 | 10 | 3.90302 | 3,773 | 3,611 |
| NZ_CP013727 | MBE1283 | 35 | 22 | 13 | 12 | 3.97993 | 3,856 | 3,681 |
| NZ_CP014700 | S499 | 33 | 19 | 14 | 11 | 3.93593 | 3,819 | 3,671 |
| NZ_CP014783 | B15 | 29 | 19 | 10 | 10 | 4.00675 | 3,875 | 3,704 |
| NZ_CP016913 | RD7-7 | 31 | 17 | 14 | 8 | 3.68821 | 3,656 | 3,483 |
| NZ_DF836091 | CMW1 | 30 | 20 | 10 | 11 | 3.90857 | 3,901 | 3,706 |
| NZ_JCOC01000001 | EBL11 | 35 | 23 | 12 | 11 | 3.92932 | 3,773 | 3,682 |
| NZ_JMEG01000001 | B1895 | 38 | 24 | 14 | 12 | 4.10728 | 4,026 | 3,623 |
| NZ_JQNZ01000001 | X1 | 40 | 28 | 12 | 10 | 3.9211 | 3,766 | 3,619 |
| NZ_JTJG01000001 | JJC33M | 36 | 25 | 11 | 12 | 3.96166 | 3,952 | 3,796 |
| NZ_JXAT01000001 | LPL-K103 | 36 | 23 | 13 | 9 | 3.87327 | 3,743 | 3,637 |
| NZ_JZDI01000001 | 12B | 69 | 49 | 20 | 11 | 7.59676 | 8,194 | 7,985 |
| NZ_KB206086 | DC-12 | 28 | 19 | 9 | 11 | 4.01656 | 3,984 | 3,842 |
| NZ_KN723307 | TF281 | 31 | 20 | 11 | 11 | 3.98764 | 3,782 | 3,571 |
| NZ_LGYP01000001 | 629 | 31 | 18 | 13 | 10 | 3.90337 | 3,785 | 3,427 |
| NZ_LJAU01000001 | Bs006 | 45 | 30 | 15 | 10 | 4.17309 | 4,074 | 3,969 |
| NZ_LJDI01000020 | XK-4-1 | 37 | 24 | 13 | 12 | 3.94181 | 3,821 | 3,701 |
| NZ_LMAG01000001 | RHNK22 | 38 | 27 | 11 | 12 | 3.97818 | 3,837 | 3,698 |
| NZ_LMAT01000001 | Jxnuwx-1 | 40 | 27 | 13 | 10 | 4.08932 | 4,008 | 3,870 |
| NZ_LMUC01000016 | H57 | 34 | 23 | 11 | 11 | 3.95883 | 3,859 | 3,732 |
| NZ_LPUP01000011 | 11B91 | 33 | 20 | 13 | 10 | 4.02366 | 3,892 | 3,702 |
| NZ_LQQW01000001 | M49 | 41 | 30 | 11 | 11 | 3.88665 | 3,741 | 3,617 |
| NZ_LQYO01000001 | B4140 | 39 | 25 | 14 | 11 | 4.01425 | 3,847 | 3,713 |

(Continued)

| Accession | Strains | Biosynthetic gene clusters | | | | Genome of strains | | |
|---|---|---|---|---|---|---|---|---|
| | | Total | Known | Unknown | Type | Size (Mb) | Gene | Protein |
| NZ_LQYP01000001 | B425 | 29 | 20 | 9 | 9 | 3.9682 | 4,034 | 3,844 |
| NZ_LYUG01000001 | SRCM101266 | 31 | 19 | 12 | 11 | 3.76536 | 3,781 | 3,628 |
| NZ_LZZO01000001 | SRCM101294 | 32 | 20 | 12 | 10 | 3.96275 | 3,982 | 3,850 |

**Note:**
"Total" of Biosynthesis gene clusters includes "Known" and "Unknown." "Known" of Biosynthesis gene clusters is inferred from the MIBiG (Minimum Information about a Biosynthetic Gene cluster, http://mibig.secondarymetabolites.org). "Unknown" of Biosynthesis gene clusters is detected by Cluster Finder and further categorized into putative ("Cf_putative") biosynthetic types. A full integration of the recently published Cluster Finder algorithm now allows the use of this probabilistic algorithm to detect putative gene clusters of unknown types; "–" of host is unrecorded.

**Table 3 Biosynthetic gene clusters of Y2(NC_017912) strain.**

| Cluster | Type | Most similar known cluster | MIBiG BGC-ID |
|---|---|---|---|
| Cluster 1 | Nrps | Surfactin_biosynthetic_gene_cluster (43% of genes show similarity) | BGC0000433_c1 |
| Cluster 2 | Cf_putative | – | – |
| Cluster 3 | Cf_putative | – | – |
| Cluster 4 | Cf_fatty_acid | – | – |
| Cluster 5 | Phosphonate | Pactamycin_biosynthetic_gene_cluster (3% of genes show similarity) | BGC0000119_c1 |
| Cluster 6 | Cf_saccharide | Plantathiazolicin/plantazolicin_biosynthetic_gene_cluster (33% of genes show similarity) | BGC0000569_c1 |
| Cluster 7 | Cf_putative | – | – |
| Cluster 8 | Otherks | – | – |
| Cluster 9 | Cf_fatty_acid | – | – |
| Cluster 10 | Cf_putative | – | – |
| Cluster 11 | Terpene | – | – |
| Cluster 12 | Cf_fatty_acid | – | – |
| Cluster 13 | Cf_putative | – | – |
| Cluster 14 | Cf_putative | – | – |
| Cluster 15 | Transatpks | Macrolactin_biosynthetic_gene_cluster (90% of genes show similarity) | BGC0000181_c1 |
| Cluster 16 | Nrps-Transatpks | Bacillaene_biosynthetic_gene_cluster (85% of genes show similarity) | BGC0001089_c1 |
| Cluster 17 | Nrps-Transatpks | Fengycin_biosynthetic_gene_cluster (93% of genes show similarity) | BGC0001095_c1 |
| Cluster 18 | Terpene | – | – |
| Cluster 19 | Cf_saccharide-T3pks | – | – |
| Cluster 20 | Transatpks | Difficidin_biosynthetic_gene_cluster (100% of genes show similarity) | BGC0000176_c1 |
| Cluster 21 | Cf_putative | – | – |
| Cluster 22 | Nrps-Bacteriocin | Bacillibactin_biosynthetic_gene_cluster (100% of genes show similarity) | BGC0000309_c1 |
| Cluster 23 | Cf_saccharide | – | – |
| Cluster 24 | Nrps | – | – |
| Cluster 25 | Cf_saccharide | Teichuronic_acid_biosynthetic_gene_cluster (100% of genes show similarity) | BGC0000868_c1 |
| Cluster 26 | Cf_putative | – | – |
| Cluster 27 | Cf_saccharide | Bacilysin_biosynthetic_gene_cluster (100% of genes show similarity) | BGC0001184_c1 |
| Cluster 28 | Cf_putative | – | – |
| Cluster 29 | Lantipeptide | Mersacidin_biosynthetic_gene_cluster (90% of genes show similarity) | BGC0000527_c1 |
| Cluster 30 | Cf_saccharide | – | – |
| Cluster 31 | Cf_putative | – | – |

**Note:**
"Cf putative" refers to putative biosynthetic types (unknown types) detected by Cluster Finder and further categorized, known types are from the MIBiG (Minimum Information about a Biosynthetic Gene cluster, http://mibig.secondarymetabolites.org).

Select Gene Cluster:

Overview 1 2 3 4 5 6 7 8 9 10 11 12 13 14 15 16 17 18 19 20 21 22 23 24 25 26 27 28 29 30 31

**Identified secondary metabolite clusters**

| Cluster | Type | From | To | Most similar known cluster | MIBiG BGC-ID |
|---|---|---|---|---|---|
| The following clusters are from record NC_017912.1: | | | | | |
| Cluster 1 | Nrps | 338057 | 358115 | Surfactin_biosynthetic_gene_cluster (43% of genes show similarity) | BGC0000433_c1 |
| Cluster 2 | Cf_putative | 382926 | 400100 | - | - |
| Cluster 3 | Cf_putative | 457505 | 463879 | - | - |
| Cluster 4 | Cf_fatty_acid | 553834 | 571410 | - | - |
| Cluster 5 | Phosphonate | 611959 | 652849 | Pactamycin_biosynthetic_gene_cluster (3% of genes show similarity) | BGC0000119_c1 |
| Cluster 6 | Cf_saccharide | 707404 | 724559 | Plantathiazolicin_/_plantazolicin_biosynthetic_gene_cluster (33% of genes show similarity) | BGC0000569_c1 |
| Cluster 7 | Cf_putative | 815908 | 837072 | - | - |
| Cluster 8 | Otherks | 928506 | 969750 | - | - |
| Cluster 9 | Cf_fatty_acid | 997814 | 1018791 | - | - |
| Cluster 10 | Cf_putative | 1019821 | 1028475 | - | - |
| Cluster 11 | Terpene | 1052066 | 1072806 | - | - |
| Cluster 12 | Cf_fatty_acid | 1099917 | 1105735 | - | - |
| Cluster 13 | Cf_putative | 1132407 | 1140612 | - | - |
| Cluster 14 | Cf_putative | 1217728 | 1234302 | - | - |
| Cluster 15 | Transatpks | 1414044 | 1469984 | Macrolactin_biosynthetic_gene_cluster (90% of genes show similarity) | BGC0000181_c1 |
| Cluster 16 | Nrps-Transatpks | 1766710 | 1842979 | Bacillaene_biosynthetic_gene_cluster (85% of genes show similarity) | BGC0001089_c1 |
| Cluster 17 | Nrps-Transatpks | 2075376 | 2178857 | Fengycin_biosynthetic_gene_cluster (93% of genes show similarity) | BGC0001095_c1 |
| Cluster 18 | Terpene | 2224424 | 2246307 | - | - |
| Cluster 19 | Cf_saccharide-T3pks | 2296276 | 2344896 | - | - |
| Cluster 20 | Transatpks | 2506980 | 2581912 | Difficidin_biosynthetic_gene_cluster (100% of genes show similarity) | BGC0000176_c1 |
| Cluster 21 | Cf_putative | 3239477 | 3252309 | - | - |
| Cluster 22 | Nrps-Bacteriocin | 3278880 | 3345673 | Bacillibactin_biosynthetic_gene_cluster (100% of genes show similarity) | BGC0000309_c1 |
| Cluster 23 | Cf_saccharide | 3533963 | 3550378 | - | - |
| Cluster 24 | Nrps | 3624364 | 3661226 | - | - |
| Cluster 25 | Cf_saccharide | 3683737 | 3738552 | Teichuronic_acid_biosynthetic_gene_cluster (100% of genes show similarity) | BGC0000868_c1 |
| Cluster 26 | Cf_putative | 3771770 | 3782210 | - | - |
| Cluster 27 | Cf_saccharide | 3882481 | 3944920 | Bacilysin_biosynthetic_gene_cluster (100% of genes show similarity) | BGC0001184_c1 |
| Cluster 28 | Cf_putative | 4020100 | 4026182 | - | - |
| Cluster 29 | Lantipeptide | 4078161 | 4102145 | Mersacidin_biosynthetic_gene_cluster (90% of genes show similarity) | BGC0000527_c1 |
| Cluster 30 | Cf_saccharide | 4170395 | 4195268 | - | - |
| Cluster 31 | Cf_putative | 4203706 | 4219124 | - | - |

**Figure 5** Screenshot of visualization of the total number of biosynthetic gene clusters of the Y2 strain.

across different machines (*Boettiger, 2015*). Dockerfiles can maintain and update related adjustments, rapidly recover from system failure events, control versions, and build application environments with the optimal flexibility. BGDMdocker Images enable portability and modular reuse. Bioinformatics tools are written in a variety of languages and require different operating environment configurations across platforms. Docker technology is capable of executing the same functions and services in different environments without additional configurations (*Folarin, Dobson & Newhouse, 2015*), thus creating reproducible tools with high efficiency. By constructing pipelines with different tools, bioinformaticians may automatically and effectively analyze biological problems of interest. The BGDMdocker Container enables application isolation with high efficiency and flexibility. Applications may run Container independently with Docker technology, and each management command (start, stop, boot, etc.) may be executed in seconds or milliseconds. Hundreds or thousands of Containers may be run on a single host at same time (*Ali, El-Kalioby & Abouelhoda, 2016*), thus ensuring that the failure of

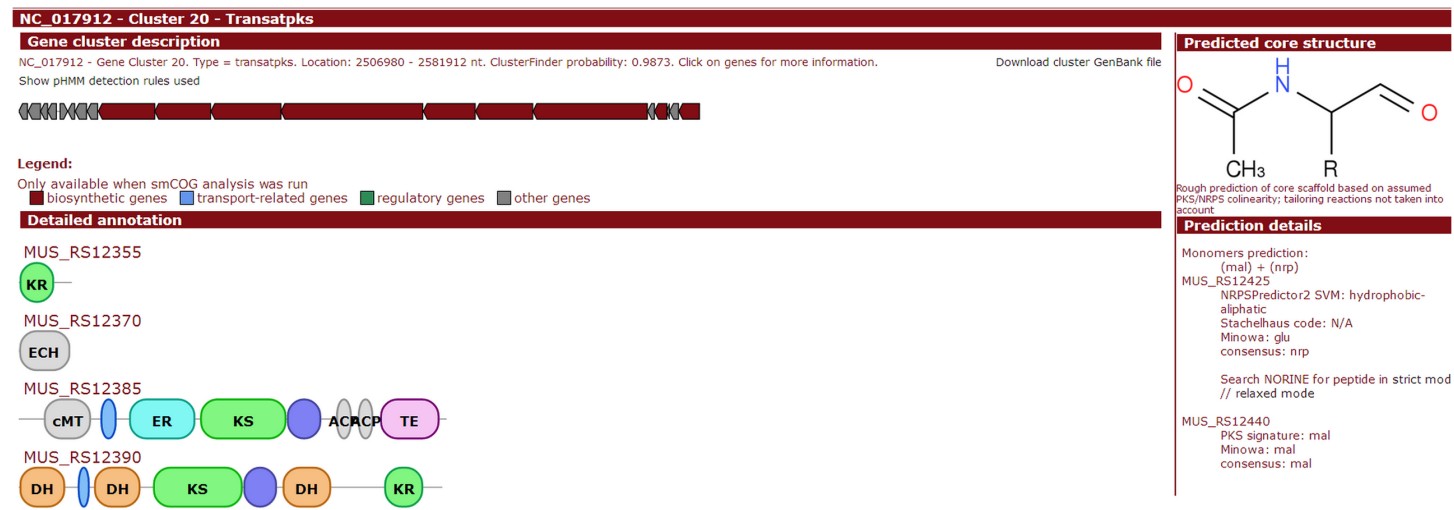

**Figure 6** Screenshot of website (http://bapgd.hygenomics.com/pangenome/home) for the visualization of Transatpks type cluster of Y2 (most similar known cluster to the Difficidin_biosynthetic_gene_cluster).

Table 4 Function of BGDMdocker workflow compared with other pan-genome tools.

| Tools | Automatic installation | Cross platform | Result visualization | Genome annotation | Gene cluster mining |
|---|---|---|---|---|---|
| Roary | × | × | × | √ | × |
| PGPA | × | × | × | × | × |
| SplitMEM | × | × | × | × | × |
| PanGP | × | × | × | × | × |
| PanTools | × | × | √ | × | × |
| BGDMdocker | √ | √ | √ | √ | √ |

**Note:**
√ is provided with the function, × is not provided with the function.

one task does not cause disruption of the entire process: new Containers may be initialized rapidly to continue the task until the completion of the entire process, thus improving overall efficiency.

In recent years, several online tools and software suites have been developed for pan-genome analysis, including Roary, PGPA, SplitMEM, PanGP, and PanTools. However, generally, the installation of these pipelines with many dependencies, but a single function, is complex and challenging. Therefore, limiting researchers' ability directly focus on their analyses of interest (Table 4). Although the BGDMdocker workflow includes several tools, installing and running the software is quite simple. Biologists may automatically install, configure, and test the scripts, making these processes faster and the results repeatable.

## CONCLUSION

Here, we present a BGDMdocker workflow to achieve bacterial and viral genome annotation, pan-genome analysis, mining of biosynthetic gene clusters, and

visualization of results on a local host or online. This allows researchers to browse information for every gene, including duplication, diversity, indel events, and sequence alignments, as well as for biosynthetic gene clusters, including structure, type, description, detailed annotation, and predicted core structure of the target compounds. These tools and their installation commands and dependencies were all written in a Dockerfile. We used this Dockerfile to build a Docker Image and run Container for analyzing the pan-genome of 44 *B. amyloliquefaciens* strains retrieved from a public database. The pan-genome included a total of 172,388 genes and 2,306 core gene clusters. The visualization of the pan-genomic data included alignments, phylogenetic trees with mutations within each cluster mapped to the branches of the tree, and inference of gene losses and gains on the core-genome phylogeny for each gene cluster. In addition, 997 known (MIBiG: http://mibig.secondarymetabolites.org database) and 553 unknown (antiSMASH-predicted clusters and Pfam database) genes in biosynthetic gene clusters and orthologous groups were identified in all strains. The BGDMdocker workflow for the analysis and visualization of pan-genomes and biosynthetic gene clusters may be fully reused immediately across different computing platforms (Linux, Windows, Mac, and cloud-based systems), with flexible and rapid deployment of integrated software packages across various platforms. This workflow may also be used for other pan-genome analyses and visualization of other species. Additionally, the visual display of data provided in this study may be completely duplicated. All resulting data and relevant tools and files may be downloaded from our website (http://bapgd.hygenomics.com/pangenome/home) with no registration required.

## ACKNOWLEDGEMENTS

We thank Yilei Wu, Chao Chen, Wei Ding, and the reviewers for usability testing and valuable suggestions.

### Funding

This work was supported by the National Key R&D Program of China (2017YFD0600103), Fundamental Research Fund for Central Non-profit Scientific Institution (Y2016PT38), and CAAS-ASTIP. The funders had no role in study design, data collection and analysis, decision to publish, or preparation of the manuscript.

### Grant Disclosures

The following grant information was disclosed by the authors:
National Key R&D Program of China: 2017YFD0600103.
Fundamental Research Fund for Central Non-profit Scientific Institution: Y2016PT38.
CAAS-ASTIP.

## Competing Interests

The authors declare that they have no competing interests.

## Author Contributions

- Gong Cheng conceived and designed the experiments, performed the experiments, analyzed the data, contributed reagents/materials/analysis tools, wrote the paper, prepared figures and/or tables, reviewed drafts of the paper.
- Quan Lu conceived and designed the experiments, performed the experiments, contributed reagents/materials/analysis tools, reviewed drafts of the paper.
- Ling Ma conceived and designed the experiments, contributed reagents/materials/analysis tools, reviewed drafts of the paper.
- Guocai Zhang contributed reagents/materials/analysis tools, reviewed drafts of the paper.
- Liang Xu reviewed drafts of the paper.
- Zongshan Zhou conceived and designed the experiments, contributed reagents/materials/analysis tools, reviewed drafts of the paper.

## Data Availability

*Bacillus amyloliquefaciens* pan-genome Database: http://bapgd.hygenomics.com/pangenome/home

DockerHub:

https://hub.docker.com/r/cgwyx/bgdmdocker/

GitHub:

https://github.com/cgwyx/BGDMdocker

cheng, wyx (2017): BGDMdocker.figshare.

https://doi.org/10.6084/m9.figshare.5122363.v1

## Supplemental Information

Supplemental information for this article can be found online at http://dx.doi.org/10.7717/peerj.3948#supplemental-information.

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
