# Peer review of "BGDMdocker: a Docker workflow for data mining and visualization of bacterial pan-genomes and biosynthetic gene clusters"

_PeerJ, doi:10.7717/peerj.3948_

## Round 0.1 · original submission · Minor Revisions

Both reviewers found the work interesting and important. The comments are very minor.

·

Basic reporting

The manuscript submitted by Cheng et al. describes BGDM Docker - a tool specifically designed to evaluate bacterial pan-genomes. BGDM Docker integrates a software for genome annotation (Prokka), a software for pan-genome analysis and classification (PanX) and a software dedicated to identify clusters of biosynthetic genes, along with the metabolites that can be produced from such clusters (anti-SMASH). These software are implemented in a single docker container, along with all dependencies necessary for deployment and use in different computing environments. Overall, BGDM Docker represents an interesting tool for comparative genomics analyses and its implementation in a docker platform helps to facilitate installation and ensure reproducibility of data analyses across laboratories.

The manuscript is sound and adequately written, highlighting several important aspects regarding the currently available tools for pan-genome analysis and the contribution of docker-based systems to the development of bioinformatics.

There are a few points, however, that must be checked by the authors:

- The software is available for download from the manuscript´s website and also from github. However, it would be interesting if authors also made it available from Dockerhub (https://hub.docker.com/), since, in this case, the functional docker image would be directly available for testing and download, through docker daemon.

- Several references, cited along the text, are not present in the manuscript´s reference list.

- Some links in the manuscript´s webpage do not seem to be working properly (such as http://42.96.173.25:8000/bamf_gbk44, for example).

Experimental design

Authors provide an adequate description of the software capabilities, as well as detailed instructions for its implementation and use. However, these instructions suggest that users implement the three software available in the BGDMdocker container (Prokka, PanX and anti-SMASH) independently. Wouldn´t it be simpler (particularly to unexperienced users) to integrate them though a single script command (pipeline)?

Moreover, as a suggestion, it would be interesting to provide in the instructions an option to run BGDMdocker without having to enter the container, which should facilitate integration of this software with external pipeline-building tools like Snakemake and Luigi, among others. One option would be to provide the command/script in the Docker CMD option, so the end user can execute the pipeline through a command similar to the following:

$ sudo docker run -it BGDMdocker $pipelinecommand $filein

This command could be called in Snakemake, for example, integrating the entire workflow.

Validity of the findings

BGDMdocker provides interesting results and has advantages when compared to many of the currently existing tools for pan-genome analysis. However, the data provided through the manuscript´s website is difficult to download, since the files ad up to more than 600 Mb each, probably due to the presence of large fasta files, which are not relevant for a reader that only wishes to evaluate the software´s capabilities. Thus, while this is not a major point in this critique, I would recommend that authors provide the more relevant output files, such as images, tables and html pages separately in the github or figshare repositories.

Additional comments

The manuscript describes a worthwhile tool for pan-genome analyses of bacterial genomes. Its implementation in a docker container shall facilitate its distribution, installation and use in different computing environments, contributing to generate reproducible results across laboratories. The manuscript is sound and can be considerably improved with a few modifications, such as the ones described above.

Reviewer 2 ·

Basic reporting

The English writing is clear.
More background of Docker should be given.
The data and software of the manuscript is public available.

Experimental design

The research idea is novel, which introduces BDGMdockter to be utilized in biomedical research. Detailed instructions to implement BDGMdockter are also given.

Validity of the findings

No comment. Please check General comments.

Additional comments

This work describes a pipeline “BDGMdockter” for analysis and visualization of bacterial pan-genomes and biosynthetic gene clusters. The “BDGMdockter” could be used in PC and distributed in cloud service, which might be useful in biomedical research.

1. github is also a popular free platform for software repository. What is the advantage of Docker compared to github?
2. BDGMdockter is basically a wrapper for existing software Prokka, panX, and antiSMASH. However, the utilization of BDGMdockter seems complicated for biologists to understand with too many command lines and needs high privileges of computer such as “sudo”. Is there any way to use BDGMdockter without root privilege? It would be nice if more docker background could be introduced and the command lines of BDGMdockter could be simplified.
3. In the RESULT section, the authors claim “Fast and reproducible building of the BGDMdocker workflow across computing platforms using Docker” line 95. However, I did not see how “fast” BGDMdocker is and how “reproducible” BGDMdocker is. Maybe it is a good idea to show the running time and memory need for the software, and produce the results which have been already reported by other literature.
4. Line 126. The author claim “a website was built for the interactive exploration of the B. amyloliquefaciens pan-genome and biosynthetic gene clusters using the BGDMdocker workflow”. However, I did not see how to upload the bacterial sequences, submit the job and produce the Table 1, Figures 2 &3 from the website http://pangenome.zggskj.com/home if the interactive exploration is what it means. A detailed steps should be given for interactive exploration.

---

## Round 0.2 · Minor Revisions

Thank you for addressing the reviewers' comments. Please further polish the language, paying special attention to the wording and punctuation highlighted in yellow in the attached file.

Reviewer 2 ·

Basic reporting

No comments

Experimental design

No comments

Validity of the findings

No comment

Additional comments

The authors address my comments.

---

## Round 0.3 · accepted · Accept

The manuscript is suitable for publication now.